# Logical Anomaly Detection with Masked Image Modeling

**Shunsuke Sakai**                                    *sshunsuke0102@gmail.com*
*Department of Engineering*
*University of Fukui*

**Tatsuhito Hasegawa**[*]                             *t-hase@u-fukui.ac.jp*
*Department of Engineering*
*University of Fukui*

**Makoto Koshino**[*]                                 *koshino@ishikawa-nct.ac.jp*
*Department of Electronics and Information Engineering*
*National Institute of Technology, Ishikawa College*

**Reviewed on OpenReview:** *https://openreview.net/forum?id=uuuaRCMYE3*

## Abstract

Detecting anomalies such as an incorrect combination of objects or deviations in their positions is a challenging problem in unsupervised anomaly detection (AD). Since conventional AD methods mainly focus on local patterns of normal images, they struggle with detecting logical anomalies that appear in the global patterns. To effectively detect these challenging logical anomalies, we introduce **L**ogical **A**nomaly **D**etection with **M**asked **I**mage **M**odeling (**LADMIM**), a novel unsupervised AD framework that harnesses the power of masked image modeling and discrete representation learning. Our core insight is that predicting the missing region forces the model to learn the long-range dependencies between patches. Specifically, we formulate AD as a mask completion task, which predicts the distribution of discrete latents in the masked region. As a distribution of discrete latents is invariant to the low-level variance in the pixel space, the model can desirably focus on the logical dependencies in the image, which improves accuracy in the logical AD. We evaluate the AD performance on five benchmarks and show that our approach achieves compatible performance without any pre-trained segmentation models. We also conduct comprehensive experiments to reveal the key factors that influence logical AD performance. Code is available at: `https://github.com/SkyShunsuke/LADMIM`.

## 1 Introduction

Automated visual inspection plays a key role in industrial applications, ensuring quality and reliability while reducing human errors and improving efficiency. Anomaly detection (AD) has recently attracted considerable attention as a promising approach to building high-performance visual inspection systems (Bergmann et al., 2019; Bao et al., 2024). At the heart of AD lies the philosophy that anomalies can be detected by capturing the concept of normality. Since anomalies are characterized by their rare occurrence and diversity, an unsupervised AD that relies solely on normal samples without requiring any anomalies has become a central paradigm in the field of AD (Liu et al., 2024).

In industrial images, defects can be classified into structural and logical anomalies. Structural anomalies refer to deviations from normal in the local features of the image, such as scratches or stains on manufactured products. On the other hand, logical anomalies are deviations from normal in the global features of the image, such as misalignment of objects or incorrect combinations of objects. Logical anomalies differ from those in normal data regarding the relationships between local features. They are more likely to occur in problem settings where multiple objects are presented in the image. Traditional benchmarks for industrial,

---

[*]Corresponding Author.

such as MVTec AD (Bergmann et al., 2019) and VisA (Zou et al., 2022), have mainly addressed structural anomalies in images containing a single object. In contrast, the newly released benchmark MVTec LOCO (Bergmann et al., 2022) focuses on problems where multiple objects are presented in the image, allowing the evaluation of detection performance for both structural and logical anomalies. In practical applications, detecting logical anomalies is often required; therefore, methods must be developed to detect both logical and structural anomalies.

Current mainstream approaches use deep convolutional neural networks to extract useful features to detect anomalies. Such features are robust against noise, object rotation, and translation and provide rich representations of normal data (Cohen & Hoshen, 2020; Roth et al., 2021; Batzner et al., 2023). In particular, representations acquired through supervised pre-training on large, curated datasets like ImageNet have high discriminative power against anomalies without any fine-tuning (Cohen & Hoshen, 2020; Roth et al., 2021). With the use of such pre-trained models, there has been a significant improvement in detection performance for structural anomalies (Roth et al., 2021; Yu et al., 2021; Lu et al., 2023). On the other hand, these methods may have lower detection performance for logical anomalies (Bergmann et al., 2022). Logical anomalies may not appear in the local features themselves but in the relationships between local features, and current mainstream approaches consider only local features (Bergmann et al., 2022).

In this study, we introduce self-supervised learning on normal images using Masked Image Modeling (MIM) to learn the relationships among local features of normal images. In MIM, part of the input image is masked, and the model is trained to restore the features in the masked region. To restore the masked region, it is necessary to understand the relationships among features in normal images. Therefore, MIM can effectively learn dependencies among local features. However, when simply predicting the pixel-level image features of the masked region, there is a problem of blurred reconstruction. This issue fundamentally arises from the uncertainty of feature positions in the masked region. To mitigate this problem, we propose to predict the probability distribution of discrete latent variables in the masked region. The probability distribution of discrete latent variables represents the composition of visual features within the masked region and is invariant to the positions of the features.

This study aims to improve detection performance for both logical and structural anomalies with masked image modeling. The contributions of this study are as follows:

- We propose a novel framework, LADMIM, which has a ViT-based architecture and leverages MIM and Hierarchical Vector Quantized Transformer (HVQ-Trans) (Lu et al., 2023) for detecting logical and structural anomalies, respectively. LADMIM mitigates the issue of positional uncertainty in AD by predicting the probability distribution of discrete latent variables for the target.

- We verify our framework on MVTecLOCO (Bergmann et al., 2022) and MVTecAD (Bergmann et al., 2019), achieving state-of-the-art detection accuracy compared with conventional MIM-based methods.

- Since MIM-based AD essentially avoids the need for tuning information bottlenecks, we demonstrate that our proposed LADMIM is amore robust approach compared with conventional non-MIM-based AD (Fig. 3), thereby LADMIM can be easily adapted to more complex logical AD problems.

Our proposed framework represents a departure from existing paradigms, paving the way for a new line of research that has the potential to establish a more robust and unified framework through future developments.

## 2 Related works

### 2.1 Unsupervised Logical Anomaly Detection

Logical anomalies are deviations in logical relationships of objects in normal images, which are different from traditional structural anomalies such as cracks and bends (Bergmann et al., 2022). Detecting such logical anomalies requires both object-centric representation and global-context modeling, which makes it challenging. Previous unsupervised AD literatures attempt to improve detection accuracy by enhancing

either of these two aspects or both. Current predominant approaches to detecting logical anomalies are categorized into (a) reconstruction-based, (b) memorybank-based. In Fig. 1, we provide a visualization of key concepts of these approaches.

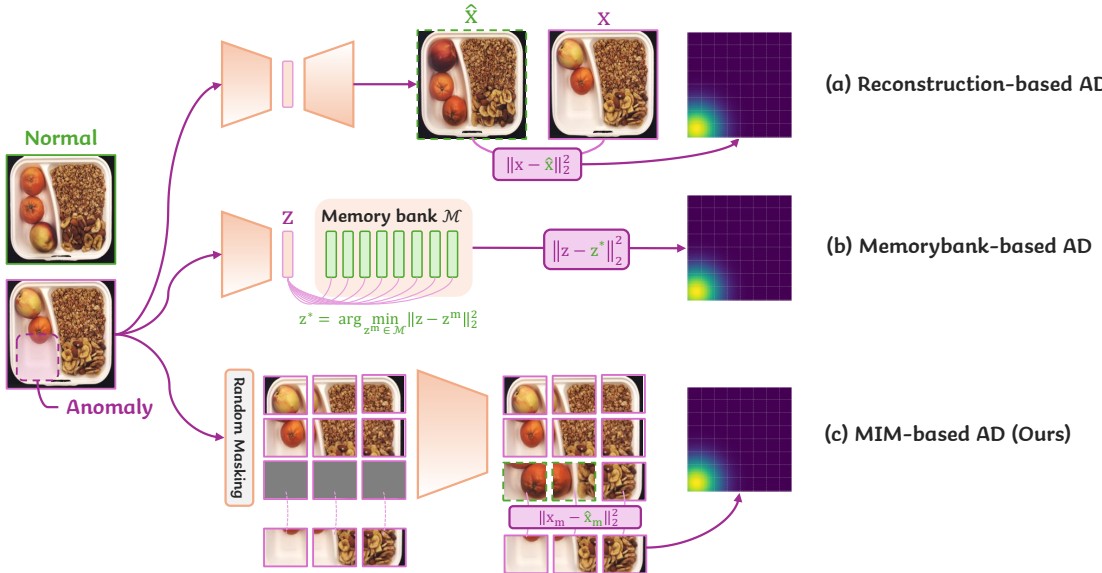

Figure 1: **Conceptual comparison** of different approaches in AD.

Reconstruction-based AD Batzner et al. (2023); Yao et al. (2023; 2024); Sugawara & Imamura (2024); Lu et al. (2023) learns to reconstruct only normal images using autoencoders (Fig. 1a). Because of the poor generalizability of the model for reconstructing anomalous features, these anomalous features are ideally reconstructed to corresponding normal features, enabling reconstruction error as a reasonable anomaly scoring metric. Although these reconstruction-based AD methods can detect certain logical anomalies by using a spatial bottleneck structure, they inevitably increase the reconstruction error for normal images Batzner et al. (2023); Lu et al. (2023). Additionally, in the absence of a low-dimensional information bottleneck, the model tends to learn an identity mapping, which conveys little information about anomalies while being globally optimal with respect to the reconstruction objective. This shortcoming of reconstruction-based methods is known as the Identity-shortcut (ID-shortcut) Batzner et al. (2023); Lu et al. (2023).

Memorybank-based methods Liu et al. (2023b); Kim et al. (2024); Roth et al. (2021) store features of normal images in a storage called a memorybank. During inference, these methods detect anomalies by measuring the distance between the input image's features and those stored in the memorybank. Some methods attempt to detect logical anomalies with memorybank and achieve competitive accuracy on public benchmarks Liu et al. (2023b); Kim et al. (2024). However, these lines of work require pre-trained segmentation networks to extract object-centric representations, thereby constraining their applicability and generalization capability.

## 2.2 Anomaly Detection with Masked Image Modeling

Masked Image Modeling (MIM) is a self-supervised learning framework that involves masking a portion of the input image and learning to predict the features of the masked region from the visible regions. MIM-based AD (Fig. 1c)is a promising approach for learning the logical relationships between objects and features in normal images. However, conventional methods (Huang et al., 2023a; Li et al., 2020; Yan et al., 2021; Schwartz et al., 2024; Yao et al., 2022; Yang et al., 2023) struggle with low detection performance compared to other reconstruction-based and memorybank-based methods. A critical challenge of MIM is the uncertainty in masked regions, which leads to undesirable prediction errors in normal regions. To mitigate this issue, we propose predicting the probability distribution of discrete latents in the masked regions, which is invariant to the order of the latents. Additionally, we employ different models to detect logical and structural

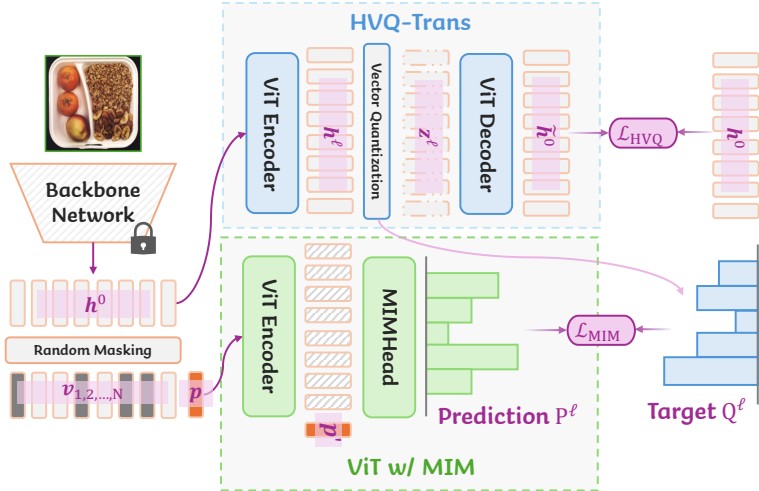

Figure 2: **Overview of our framework.** Our framework consists of two main components: HVQ-Trans and ViT trained with the MIM objective.

anomalies, respectively. In summary, our proposed AD framework can ideally capture the challenging logical relationship of normal images, which in turn contributes to more accurate detection for both logical and structural anomalies.

## 3 Methodology

### 3.1 Overview of the Proposed Framework

An overview of the proposed framework is shown in Fig. 2. The proposed framework detects structural and logical anomalies using different mechanisms. We call this framework for both logical and structural AD **L**ogical **A**nomaly **D**etection with **M**asked **I**mage **M**odeling (**LADMIM**). Structural anomalies are detected using the reconstruction error of features by HVQ-Trans Lu et al. (2023), which also serves as the tokenizer. Logical anomalies are detected through the prediction error of the probability distribution predicted by the ViT-based model, which is self-supervised via MIM using the probability distribution of discrete latent variables from HVQ-Trans as the prediction target. The anomaly scores calculated by each mechanism are then integrated to compute the final anomaly score of the input image. The normality or anomalousity of an image is determined by applying an appropriate threshold to the anomaly score.

Compared to reconstruction-based methods Batzner et al. (2023); Yao et al. (2023; 2024), the proposed framework does not suffer from the ID-shortcut of the input because identity mapping results in large reconstruction errors in the masked region. As discussed in Section 4.4, since the ID-shortcut does not occur in our framework, LADMIM can be scaled to a larger model.

Anomalies are detected through the interaction of more abstract visual features by running predictions in the discrete latent space. Moreover, when no masking is applied to the input image, the proposed framework can be considered a distillation-based method, where HVQ-Trans acts as the teacher and ViT as the student. Masking the input image in the proposed framework promotes learning of long-range dependencies among features in normal images, compared to distillation-based methods Batzner et al. (2023); Guo et al. (2023); Zhang et al. (2023). Compared to memory bank-based methods Liu et al. (2023b); Kim et al. (2024), the proposed framework shares similarities with PSAD Kim et al. (2024) and ComAD Liu et al. (2023b) in using the area distribution of objects to represent normal features. While these methods Kim et al. (2024); Liu et al. (2023b) require the separate preparation of a segmentation model, the proposed framework utilizes the discrete latent variables of HVQ-Trans, acquired through fully unsupervised learning via the image reconstruction task.

The novelty of the proposed framework lies in (i) leveraging MIM prediction errors to detect logical anomalies and (ii) using the probability distribution (histogram) of discrete latent variables within the masked region as the MIM prediction target, rather than predicting position-wise tokens. To realize this, we employ HVQ-Trans as a unified module that simultaneously supports structural anomaly detection and tokenization: it detects structural anomalies via reconstruction error while producing a hierarchical discrete latent representation during reconstruction, which naturally serves as a tokenizer for MIM by enabling coarse-to-fine prediction at multiple granularity levels, capabilities not explicitly provided by non-hierarchical VQ variants that are not designed for structural AD. Our ViT is further designed to predict a token histogram for the masked region, explicitly addressing positional uncertainty (*e.g.,* local permutations or misalignment) and thereby facilitating the detection of position-invariant logical anomalies without relying on heuristic search procedures that can arise from an information bottleneck. This design is technically motivated by prior work combining MIM with VQ tokenizers Bao et al. (2021); Peng et al. (2022), where discrete visual tokens provide compact, easier-to-predict targets and reduce the risk of overfitting to unnecessary low-level details; building on this foundation, our histogram prediction extends MIM to a setting where robustness to positional ambiguity is essential. While a performance gap to the current SoTA remains, we believe these design choices clarify the technical significance of using MIM with discrete token distributions for anomaly detection and highlight its future potential.

## 3.2 HVQ-Trans

Hierarchical Vector Quantized Transformer (HVQ-Trans) Lu et al. (2023) is a reconstruction-based AD method incorporating a hierarchical vector quantization mechanism. HVQ-Trans learns a reconstruction model in the feature space of pre-trained models such as EfficientNet Tan & Le (2019). The reconstruction model consists of a 4-layer encoder-decoder Vision Transformer (ViT), where the output of each encoder layer is quantized, and the decoder reconstructs the features using the quantized latent variables from the encoder. The quantization of latent variables alleviates the ID shortcut problem specific to reconstruction models, achieving SoTA performance among reconstruction-based methods on MVTecAD Bergmann et al. (2019) as of 2023. Additionally, hierarchical quantization in the HVQ-Trans increases codebook usage, which is important for subsequent MIM ViT training Lu et al. (2023) Since a single large codebook in standard VQ can suffer from dead codes, especially when the number of assignments per update is small relative to codebook size, or when the encoder distribution is highly skewed. HVQ mitigates this by decomposing a large vocabulary into multiple smaller codebooks arranged in a coarse-to-fine hierarchy. Because each sub-codebook is smaller, codes are selected more frequently and receive more consistent updates, which improves per-level utilization. Moreover, the overall vocabulary corresponds to combinations across levels: if each level uses a substantial fraction of its codes, the number of reachable composite tokens increases multiplicatively with the number of levels. This yields a large effective discrete space without relying on a single massive codebook that is difficult to populate. In contrast, EMA updates van den Oord et al. (2017); Razavi et al. (2019); Zhang et al. (2024) mainly stabilize code updates but do not revive never-selected codes; reset strategies Dhariwal et al. (2020); Zhang et al. (2024) can revive dead codes but are heuristic and introduce discontinuities; and RVQ Zhang et al. (2024); Lee et al. (2022) increases capacity but may still show stage-wise imbalance in utilization and increases quantization complexity for defining MIM targets. This is a reason why we choose HVQ-Trans as a source of discrete latents among various VQ methods van den Oord et al. (2017); Peng et al. (2022); Liu et al. (2023a); Huang et al. (2023b).

In this study, HVQ-Trans is used to detect structural anomalies and simultaneously serves as a tokenizer for detecting logical anomalies. In other words, the discrete latent variables—internal representations of HVQ-Trans—are used as a teacher signal for models that detect logical anomalies.

From here, we describe the detection of structural anomalies using HVQ-Trans. First, the entire set of normal images is expressed as follows:

$$D_{\mathcal{N}} = \left\{ \mathbf{x}^{(i)} \right\}_{i=1}^{M}, \quad \mathbf{x}^{(i)} \in \mathbb{R}^{C \times H \times W}. \tag{1}$$

Here, $C$, $H$, and $W$ represent the number of channels, height, and width of the images, respectively. $M$ represents the number of training samples (*i.e.,* the number of normal images). Assuming the backbone

network is $f_\phi$, the feature extraction of a normal image $\mathbf{x}^{(i)} \in D_\mathcal{N}$ is expressed as follows:

$$\mathbf{h}^0 = f_\phi(\mathbf{x}^{(i)}), \quad \mathbf{h}^0 \in \mathbb{R}^{d_0 \times h \times w}. \tag{2}$$

Here, $d_0$, $h$, and $w$ represent the dimensions, height, and width of the feature map, respectively. HVQ-Trans reconstructs this feature map $\mathbf{h}^0$ using an encoder-decoder ViT. The feature map, flattened along the spatial dimensions, is denoted as $\mathbf{h}^{0'} \in \mathbb{R}^{d_0 \times N}$, where $N = hw$. The ViT encoder is constructed by stacking $L$ layers of standard Transformer Blocks Vaswani et al. (2017), and the features transformed by the $l$-th layer block $g_\phi^l$ are expressed as $\mathbf{h}^l = g_\phi^l(\mathbf{h}^{l-1}) \in \mathbb{R}^{d \times N}$. Here, $d$ is the embedding dimension of the ViT. The ViT decoder is similarly constructed by stacking $L$ layers of Transformer Blocks. The input to the Transformer Blocks of the decoder is the quantized output of each encoder layer. Assuming that the $l$-th layer's learnable discrete latent variable set (codebook) is $\mathbf{B}^l = \{\mathbf{b}_n^l\}_{n=1}^K$, the quantization of the final layer features of the encoder is expressed as follows:

$$\mathbf{z}^L = \mathbf{b}_i^L,$$
$$\text{where} \quad i = \operatorname*{argmin}_j \|\psi^L\left(\mathbf{h}^L\right) - \mathbf{b}_j^L\|_2^2, \quad \mathbf{b}_i^L \in \mathbb{R}^{d_q}. \tag{3}$$

The output of each encoder layer $l < L$ is quantized using the output of the encoder's final layer obtained from Eq. 3 as follows:

$$\mathbf{z}^l = \mathbf{b}_i^l,$$
$$\text{where} \quad i = \operatorname*{argmin}_j \|\psi^l\left([\mathbf{h}^{l-1}, \mathbf{z}^l]\right) - \mathbf{b}_j^l\|_2^2, \quad l \in \{1, \ldots, L-1\}. \tag{4}$$

Here, $[\cdot]$ denotes the concatenation of two vectors, and $\psi(\cdot)$ is an embedding map to the space of the same dimension as the discrete latent variables for each layer. In each layer's decoder block, the corresponding layer's discrete latent variable is used as input, and the Multi-head Self-Attention (MSA) and Multi-head Cross-Attention (MCA) mechanisms Vaswani et al. (2017) are sequentially applied. Given the quantized latent variable $\mathbf{z}^l$ of the $l$-th encoder layer, the output $\mathbf{d}^l$ of the $l$-th decoder layer is computed using the output $\mathbf{d}^{l-1}$ of the previous decoder layer as follows:

$$\mathbf{q}^l = \text{MSA}\left(\text{q} = \mathbf{W}_q\mathbf{d}^{l-1}, \text{k} = \mathbf{W}_k\mathbf{d}^{l-1}, \text{v} = \mathbf{W}_v\mathbf{d}^{l-1}\right) + \mathbf{d}^{l-1},$$
$$\tilde{\mathbf{d}}^l = \text{MCA}\left(\text{q} = \mathbf{W}_q'\mathbf{q}^l, \text{k} = \mathbf{W}_k'\mathbf{z}^l, \text{v} = \mathbf{W}_v'\mathbf{z}^l\right) + \mathbf{q}^l, \tag{5}$$
$$\mathbf{d}^l = \text{FFN}\left(\tilde{\mathbf{d}}^l\right) + \tilde{\mathbf{d}}^l.$$

MSA$(\cdot)$ and MCA$(\cdot)$ represent multi-head self-attention and multi-head cross-attention mechanisms, respectively, which take query, key, and value as inputs. FFN$(\cdot)$ represents a feed-forward network consisting of two fully connected layers and an activation function. In the multi-head cross-attention mechanism of the decoder, the output of the previous decoder layer is transformed by the weighted sum of the discrete latent variables $\mathbf{z}^l$. This process replaces the anomalous features in $\mathbf{q}^l$ with discrete latent variables, which are prototypes of normal features acquired during training, thereby increasing the reconstruction error in the anomalous areas.

During training, the input $\mathbf{h}^0$ of the encoder is reconstructed using the output $\mathbf{d}^L$ of the $L$-th layer of the decoder. That is:

$$\tilde{\mathbf{h}}^0 = \Gamma\left(\mathbf{d}^L\right), \quad \tilde{\mathbf{h}}^0 \in \mathbb{R}^{d_0 \times h \times w}, \tag{6}$$

where $\Gamma(\cdot)$ represents a mapping to the same embedding space as $\mathbf{h}^0$. The loss function used for training HVQ-Trans is defined as follows:

$$\mathcal{L}_{\text{HVQ}} \stackrel{\text{def}}{=} \|\mathbf{h}^0 - \tilde{\mathbf{h}}^0\|_2^2$$
$$+ \sum_{l=1}^L \left[\|\text{sg}\left(\mathbf{h}^l\right) - \mathbf{b}^l\|_2^2 + \|\mathbf{h}^l - \text{sg}\left(\mathbf{b}^l\right)\|_2^2\right]. \tag{7}$$

Here, sg($\cdot$) represents a stop-gradient operation that prevents gradients from being backpropagated. Since the quantization of latent variables involves non-differentiable operations, following Adiban et al. (2022); Bengio et al. (2013); van den Oord et al. (2017), this study estimates the gradient using the straight-through estimator (*i.e.,* the gradient with respect to the encoder output is replaced by the gradient with respect to the quantized encoder features).

At inference time, the anomaly score is calculated using the Mean Squared Error (MSE) between $\mathbf{h}^0$ and $\tilde{\mathbf{h}}^0$. Since vector quantization replaces continuous abnormal latents with discrete normal codes, abnormal patches in $\mathbf{h}^0$ are ideally reconstructed to corresponding normal patches. Thus, MSE is a reasonable scoring metric to detect abnormal patches. On the other hand, HVQ-Trans is a method for detecting *structural anomalies* and employs a patch-wise bottleneck structure. Therefore, it is difficult for HVQ-Trans to detect logical anomalies that require long-range feature dependencies. Next, we describe our proposed method, which focuses on detecting logical anomalies.

### 3.3 ViT for Detecting Logical Anomalies

We use a ViT-based architecture to detect logical anomalies with MIM, which contrasts with previous CNN-based MIM approaches Huang et al. (2023a); Li et al. (2020); Yan et al. (2021). A standard ViT Dosovitskiy et al. (2020) is trained on normal images using MIM, and during inference, it detects anomalies based on the prediction error of features in the masked regions. It uses the probability distribution of the discrete latent variables of HVQ-Trans for prediction. This approach mitigates the positional uncertainty of objects in the masked areas and prevents the model's predictions from being dominated by low-level features.

First, for a normal image $\mathbf{x}$, the feature map obtained from the common Backbone Network $f_\phi$ with HVQ-Trans is flattened to obtain $\mathbf{h} \in \mathbb{R}^{d_0 \times N}$. When masking a proportion $r$ ($0 < r < 1$) of the $N$ feature tokens, the positions of the masked tokens are $\mathcal{M} \subset [N]$, $|\mathcal{M}| = rN$. At this time, the input to ViT is $\{\mathbf{v}_1, \mathbf{v}_2, \ldots, \mathbf{v}_N, \mathbf{p}\}$ and:

$$\mathbf{v}_i = \begin{cases} \mathbf{h}_i & \text{if } i \notin \mathcal{M} \\ \mathbf{e} & \text{if } i \in \mathcal{M} \end{cases} \quad \text{for } i \in \{1, \ldots, N\}. \tag{8}$$

Here, $\mathbf{e}$ and $\mathbf{p}$ are learnable vectors initialized as zero vectors. ViT is composed of $L'$ layers of ordinary Transformer blocks. $\mathbf{p}$ aggregates information from visible patch tokens via self-attention and is then used to predict the distribution of discrete latents in the masked region. The output $\mathbf{p}'$ of the final layer of ViT corresponding to the prediction token is linearly mapped to predict the probability distribution of the $l$-th layer discrete latent variable of HVQ-Trans:

$$P^l = \text{MIMHead}\,(\mathbf{p}') = \text{Softmax}\,(\text{Linear}\,(\mathbf{p}'))\,, \tag{9}$$

where $P^l \in \mathbb{R}^K$ represents the probability distribution over the discrete latent variables to be predicted, which is calculated from the internal representation of HVQ-Trans. By quantizing the output of the $l$-th layer of the encoder in HVQ-Trans, the set of the indices of discrete latent variables $\mathbf{o}^l \in \mathbb{R}^N, \mathbf{o}^l_{(i)} \in [K]$ is obtained. We call these obtained indices of discrete latent variables a code prediction map. The code prediction map from the $l$-th layer of the HVQ-Trans is calculated as follows:

$$\mathbf{o}^l_{(i)} = \arg\min_j ||\psi^l([\mathbf{h}^{l-1}, \mathbf{z}^l]) - \mathbf{b}^l_j||^2_2. \tag{10}$$

The probability distribution of the discrete latent variables to be predicted is derived from $\mathbf{o}^l_{\mathcal{M}} = \{\mathbf{o}^l_{(i)} \mid i \in \mathcal{M}\}$, which is restricted to the masked region:

$$Q^l = \text{Histogram}(\mathbf{o}^l_{(i)}), \quad Q^l \in \mathbb{R}^K. \tag{11}$$

For each layer $l \in \{1, 2, ..., L\}$, with the equations 9 and 11, the loss of ViT is defined as follows:

$$\mathcal{L}_{\text{MIM}} \stackrel{\text{def}}{=} \sum_{\ell=1}^{L} \sum_{n=1}^{K} \left| P_n^l - Q_n^l \right|. \tag{12}$$

ViT is optimized with equation 12 as the objective function. By predicting features in masked regions, ViT learns the relationships among features in normal images. To mitigate the position uncertainty problem in masked regions, we employ a histogram matching loss, which is invariant to the permutation of discrete latent variables. During inference, if an area containing anomalous features not encountered during training is masked, the prediction error for that area increases.

### 3.4 Training and Inference Procedure

The proposed method is trained in two stages: (i) training HVQ-Trans, and (ii) training ViT with MIM. HVQ-Trans is trained to minimize the loss $\mathcal{L}_{\text{HVQ}}$ through a reconstruction task in the feature space. After training HVQ-Trans, ViT is trained to minimize the loss $\mathcal{L}_{\text{MIM}}$. During ViT training, the weights of HVQ-Trans are frozen and are not updated.

During inference, the reconstruction error of HVQ-Trans is evaluated using the MSE, and the average MSE across the entire image is taken as the HVQ-Trans anomaly score $S_{\text{HVQ}}$ for that image. Additionally, multiple random block-wise masks are generated, and the prediction error of the probability distribution for each mask is evaluated using the L1 distance, with the average being taken as the ViT anomaly score $S_{\text{MIM}}$. The overall anomaly score of the proposed method is obtained by summing the standardized $S_{\text{HVQ}}$ and $S_{\text{MIM}}$, denoted as $S'_{\text{HVQ}}$ and $S'_{\text{MIM}}$, respectively.

## 4 Experiments

### 4.1 Dataset and Evaluation Metrics

**MVTecLOCO** Bergmann et al. (2022) is a dataset focusing on five types of industrial products, containing more than 300 normal images for each type and more than 3600 images in total. This dataset contains five different AD tasks: BreakfastBox (Box), JuiceBottle (Bottle), Pushpins (Pins), ScrewBag (Bag), and SplicingConnectors (Cable). It primarily addresses the problem setting where multiple objects are included in an image and allows for the evaluation of detection performance for both structural and logical anomalies. Annotations (ground truth) of anomalous regions at the pixel level are provided for both logical and structural anomalies. The details of the dataset statistics are presented in Table 1. We use the training set for model training, the validation set for optimizing hyperparameters such as the number of epochs, and the test set for evaluation. We also evaluate detection performance on **MVTecAD** Bergmann et al. (2019), which is a popular benchmark for structural anomalies. MVTecAD contains over 5000 images divided into 15 categories, including objects and textures.

We employ image-level AUROC [%] as an evaluation metric to fairly compare detection performance with other methods. AUROC is calculated by plotting the false positive rate on the horizontal axis and the true positive rate on the vertical axis, dynamically adjusting the anomaly threshold $\tau$ to draw the curve. AUROC reaches its maximum when the true positive rate is 1 and the false positive rate is 0 at a certain threshold $\tau^*$, and it becomes 0.5 when using a random predictor.

Table 1: **Dataset statistics** of MVTecLOCO across different categories. The number of test images is grouped as normal/logical/structural.

| # Images | Box | Bag | Pins | Cable | Bottle | Total |
|---|---|---|---|---|---|---|
| Training | 351 | 360 | 372 | 354 | 335 | **1772** |
| Validation | 62 | 60 | 69 | 59 | 54 | **304** |
| Test | 102/83/90 | 122/137/82 | 138/91/81 | 119/108/85 | 94/142/94 | **575/561/432** |

Table 2: **Quantitative results on** the MVTecLOCO. We report image-level AUROC[%] in the single-class setting. The best results among conventional and MIM-based methods are highlighted in **bold**. Notably, our approach significantly improves AD accuracy on MIM-based approaches and surpasses conventional AD methods on average detection accuracy, narrowing the gap between current SoTA methods (EAD, PUAD).

| Category | Conventional | | | MIM | | | | SoTA | |
|---|---|---|---|---|---|---|---|---|---|
| | PC | ComAD | GCAD | SSM | SMAE | MDY | **Ours** | EAD | PUAD |
| **Logical** | | | | | | | | | |
| Box | 74.8 | **94.7** | 87.0 | 32.9 | 74.3 | 74.0 | 86.2 ($\pm$0.760) | 76.1 | 92.4 |
| Bottle | 93.9 | 90.9 | **100.0** | 59.8 | 96.8 | 57.8 | 99.0 ($\pm$0.200) | 96.3 | 99.1 |
| Pins | 63.6 | 89.0 | 97.5 | 64.2 | 73.9 | 43.7 | 73.5 ($\pm$0.695) | 98.8 | 91.7 |
| Bag | 57.8 | **79.7** | 56.0 | 60.2 | 58.0 | 47.2 | 58.1 ($\pm$3.783) | 58.3 | 77.8 |
| Cable | 84.4 | **91.9** | 89.7 | 50.3 | 76.6 | 65.0 | 88.0 ($\pm$0.836) | 94.2 | 92.3 |
| **Average** | 74.0 | **87.7** | 86.0 | 53.5 | 75.9 | 57.6 | 80.9 ($\pm$0.692) | 84.7 | 90.7 |
| **Structural** | | | | | | | | | |
| Box | 80.1 | 70.0 | 79.2 | 32.6 | 71.7 | 54.7 | **81.0** ($\pm$0.755) | 76.1 | 75.3 |
| Bottle | 98.5 | 80.5 | **99.9** | 59.8 | 89.9 | 53.4 | 98.8 ($\pm$0.286) | 96.3 | 99.2 |
| Pins | 87.9 | 93.8 | **95.1** | 51.5 | 70.9 | 85.1 | 85.9 ($\pm$1.521) | 98.8 | 98.5 |
| Bag | **92.0** | 65.0 | 87.1 | 56.2 | 63.0 | 49.2 | 86.7 ($\pm$0.817) | 58.3 | 92.0 |
| Cable | 88.0 | 63.8 | **98.3** | 53.0 | 64.5 | 59.2 | 90.0 ($\pm$1.132) | 94.2 | 93.9 |
| **Average** | 89.3 | 74.6 | **91.9** | 50.6 | 72.0 | 60.3 | 88.5 ($\pm$0.516) | 84.7 | 91.8 |
| **Overall** | 81.7 | 81.2 | 83.4 | 52.1 | 74.0 | 59.0 | **84.7** ($\pm$0.561) | 88.3 | 91.2 |

## 4.2 Implementation Details

The resolution of the input images is uniformly set to $224 \times 224$ [px]. The backbone network used in HVQ-Trans and ViT is the Patch Description Network (PDN) Batzner et al. (2023). Average pooling with a kernel size of 2 is applied to the features obtained from PDN, resulting in a feature map $\mathbf{h}_0 \in \mathbb{R}^{384 \times 24 \times 24}$. The number of layers $L$ in HVQ-Trans is set to 4, the codebook size is 512, and the codebook dimension is 64. AdamW Loshchilov & Hutter (2017) is used as the optimizer, with a weight decay of 1e-4. HVQ-Trans was trained for 1000 epochs. Training took approximately 6 hours using a CPU (Intel Core i9-14900KF), 32 GB of memory, and a GPU (NVIDIA GeForce RTX 4090 24 GB).

The number of layers $L'$ in the ViT is set to 4, and a block-wise masking strategy is adopted Bao et al. (2021); Peng et al. (2022). AdamW Loshchilov & Hutter (2017) is used as the optimizer, with a weight decay of $1 \times 10^{-6}$. The ViT was trained for 500 epochs, taking about 2 hours per class.

## 4.3 Main Experimental Results

In Table 2, we report the image-level AUROC on MVTecLOCO for our proposed method and other anomaly detection methods in the one-class setting. We show the average and standard deviation of the performance of models trained with five different random seeds. Table 2 shows the effectiveness of the proposed method compared to other anomaly detection approaches, as well as the performance improvements over existing MIM-based methods. The proposed method demonstrates performance surpassing memorybank-based methods (PatchCore Roth et al. (2021), ComAD Liu et al. (2023b)) and conventional MIM-based methods (SSM Huang et al. (2023a), SMAE Yao et al. (2022), MAEDAY Schwartz et al. (2024)). While the proposed method surpasses GCAD Bergmann et al. (2022), there exists a 3.6 % performance gap between our method and one of the current SoTA methods, EfficientAD Batzner et al. (2023), in terms of overall AUROC. However, the main limitation of EfficientAD is that it relies on the information bottleneck in the autoencoder to detect logical anomalies. Since the information of anomalous regions decays through the information bottleneck, logical anomalies can be detected by calculating the reconstruction error. However, an information bottleneck that is too tight (*i.e.,* the features have smaller dimensionality) leads to significant

errors in normal regions. One must determine the proper dimensionality of features through time-consuming hyperparameter tuning. Similarly, this also applies to PUAD Sugawara & Imamura (2024), which is an extension of EfficientAD.

In Table 3, we also report the detection performance compared with current SoTA logical anomaly detection approaches Hsieh & Lai (2024); Kim et al. (2024); Sugawara & Imamura (2024); Batzner et al. (2023); Zhang et al. (2025); Fučka et al. (2025). For a single-class setting, SoTA methods exhibit significantly better performance than our approach. However, they rely on supervision for fine-tuning the segmentation model or on hand-crafted prompt engineering. Therefore, we primarily focus on comparisons with PUAD Sugawara & Imamura (2024) and EfficientAD Batzner et al. (2023). In these comparisons, our proposed framework achieves comparable performance to PUAD in the multi-class setting[1], with only a 1.2% difference in image-level AUROC. Since changing the problem from single-class to multi-class makes the task more difficult and requires the model to be more expressive, methods that rely on information bottlenecks tend to experience a drop in detection performance. The performance of EfficientAD and PUAD significantly drops by $\approx$5%.

Because our proposed method does not rely on an information bottleneck, it can be scaled to various model sizes. This suggests that the MIM-based approach has the potential to serve as a foundational model for general-purpose anomaly detection. LADMIM and EfficientAD have different feature dimensions. For a fair comparison, in Fig. 3, we also show how detection performance changes as the feature dimension of the model increases in a multi-class setting. As the feature dimension increases, our proposed method shows improved detection performance for logical anomalies, while EfficientAD does not exhibit the same trend.

Table 4 presents the detection performance on the MVTecAD dataset. Our proposed method achieves the best performance compared to conventional MIM-based approaches. While there is a performance gap between our proposed LADMIM and current SoTA methods, it is important to note that our objective is to detect both logical and structural anomalies, rather than focusing solely on structural anomalies.

The proposed method uses HVQ-Trans to detect structural anomalies and ViT to detect logical anomalies. However, it remains unclear which specific types of anomalies each model is best suited to detect. Additionally, in the proposed method, the prediction target of MIM is set as the probability distribution of discrete latent variables, but it is also possible to set simpler pixel-level image features as the prediction target. Furthermore, the types of anomalies that can be detected are greatly influenced by the visual features represented by the discrete latent variables. Therefore, in this study, we conducted experiments to answer the following questions:

- Is it necessary to use both HVQ-Trans and ViT?
- How does the prediction target of MIM affect detection performance?
- What features do the discrete latent variables represent?

### 4.3.1 Design Ablations

The original HVQ-Trans paper Lu et al. (2023) does not evaluate its performance in detecting logical anomalies using datasets like MVTecLOCO Bergmann et al. (2022). Therefore, the structural and logical AD performance when using HVQ-Trans and ViT independently is shown in Table 5.

As a result of this evaluation, it was found that HVQ-Trans achieves high detection performance for structural anomalies, while ViT achieves high detection performance for logical anomalies. Using both models simultaneously improves overall detection performance.

The structural AD performance of ViT is low, with an AUROC of 0.68, because the discrete latent variables do not retain the characteristic information of local normal features and instead capture more abstract characteristics of objects. On the other hand, ViT achieves higher logical AD performance than HVQ-Trans.

Moreover, when HVQ-Trans and ViT are used together, their detection performance exceeds the logical AD performance of either model used individually. This indicates that HVQ-Trans and ViT complement each other in detecting different types of logical anomalies.

---

[1]The multi-class setting refers to problem sets where one model learns to handle all categories at once.

Table 3: **Comparison between current SoTA AD methods** and our proposed framework on the MVTecLOCO. We report image-level AUROC[%] in both single- and multi-class settings. "Seg. Prior" denotes whether the method uses pre-trained segmentation models.

| Category | Single-class | | | | | Multi-class | | |
|---|---|---|---|---|---|---|---|---|
| | CSAD | PSAD | SALAD | LogSAD | **Ours** | PUAD | EAD | **Ours** |
| **Logical** | | | | | | | | |
| Box | 94.4 | **100** | 99.6 | N/A | 86.2 (±0.760) | **94.2** | 77.9 | 86.5 (±0.585) |
| Bottle | 94.9 | 99.1 | **99.6** | N/A | 99.0 (±0.200) | 96.6 | 90.8 | **98.7** (±0.230) |
| Pins | 99.5 | **100** | 99.9 | N/A | 73.5 (±0.695) | **76.6** | 70.8 | 72.0 (±2.018) |
| Bag | **99.9** | 99.3 | 98.6 | N/A | 58.1 (±3.783) | **70.8** | 59.0 | 62.4 (±1.420) |
| Cable | 94.8 | 91.9 | **95.8** | N/A | 88.0 (±0.836) | **87.8** | 83.5 | 87.3 (±0.979) |
| **Average** | 96.7 | 98.1 | **98.7** | 89.3 | 80.9 (±0.692) | **85.2** | 76.4 | 81.4 (±0.553) |
| **Structural** | | | | | | | | |
| Box | **91.1** | 84.9 | 88.8 | N/A | 81.0 (±0.755) | 72.5 | 77.3 | **81.5** (±1.313) |
| Bottle | 95.6 | 98.2 | **98.9** | N/A | 98.8 (±0.286) | 95.5 | 96.7 | **98.2** (±0.141) |
| Pins | 97.8 | 89.8 | **98.3** | N/A | 85.9 (±1.521) | **97.7** | 88.0 | 86.9 (±1.102) |
| Bag | 93.2 | 95.7 | **94.7** | N/A | 86.7 (±0.817) | **88.8** | 84.3 | 88.1 (±0.753) |
| Cable | 92.2 | 89.3 | **98.6** | N/A | 90.0 (±1.132) | 82.5 | **94.0** | 89.6 (±1.298) |
| **Average** | 94.0 | 91.6 | **95.8** | 93.1 | 88.5 (±0.516) | 87.4 | 88.1 | **88.9** (±0.267) |
| **Overall** | 95.3 | 94.9 | **97.3** | 91.2 | 84.7 (±0.561) | **86.3** | 82.2 | 85.1 (±0.162) |
| Seg. Prior | ✓ | ✓ | ✓ | ✓ | ✗ | ✗ | ✗ | ✗ |

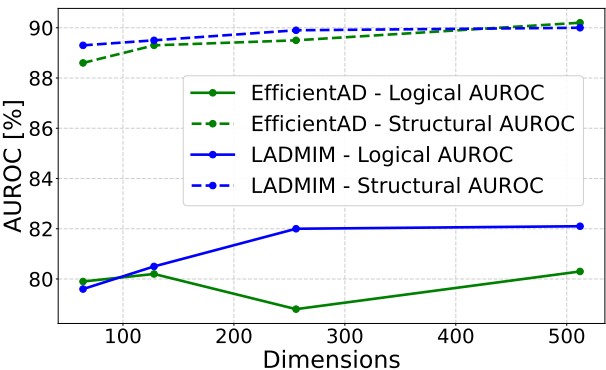

Figure 3: **Scalability analysis.** We plot the change in image-level AUROC [%] with model dimensions in the multi-class setting.

Table 4: **Performance on MVTecAD.** We report average detection AUROC [%] of LADMIM and other methods on MVTecAD Bergmann et al. (2019).

| Method | AUROC [%] |
|---|---|
| **Comparison with MIM Methods** | |
| SSM Huang et al. (2023a) | 92.0 |
| SCADN Yan et al. (2021) | 81.8 |
| LADMIM (Ours) | **93.4** |
| **Comparison with SoTA Methods** | |
| GLASS Chen et al. (2024a) | **99.9** |
| HypAD Flaborea et al. (2023) | 99.2 |
| CoMet Aqeel et al. (2025) | 99.7 |
| LADMIM (Ours) | 93.4 |

### 4.3.2 The Effect of Combining HVQ-Trans with Prior MIM-Based AD

Our proposed framework heavily relies on HVQ-Trans to detect structural anomalies. However, it remains unclear whether other MIM-based approaches can improve detection performance when combined with HVQ-Trans predictions. In Table 6, we show the average detection performance of the MIM-based methods when combined with HVQ-Trans. The results indicate that combining HVQ-Trans with MIM-based methods can enhance detection performance for both logical and structural anomalies. Although SMAE Yao et al. (2022) combined with HVQ-Trans achieves better detection performance for logical anomalies than our method, LADMIM exhibits higher overall detection performance on average.

Table 5: **Design ablation of the anomaly scoring** on MVTecLOCO. We report average image-level AUROC[%] across all categories.

| $\mathcal{S}_{\mathrm{HVQ}}$ | $\mathcal{S}_{\mathrm{MIM}}$ | SA | LA | Avg. |
|:---:|:---:|:---:|:---:|:---:|
| ✓ | | **91.2** | 76.7 | 83.3 |
| | ✓ | 68.7 | 79.3 | 73.6 |
| ✓ | ✓ | 90.3 | **83.1** | **86.6** |

Table 6: **Detection performance when combining HVQ-Trans** with MIM-based AD.

| Method | SA | LA | Avg. |
|:---|:---:|:---:|:---:|
| SMAE | 72.0 | 75.9 | 73.9 |
| SMAE [w/ HVQ] | 80.0 | **85.4** | 82.6 |
| MAEDAY | 60.3 | 57.6 | 58.9 |
| MAEDAY [w/ HVQ] | 85.4 | 71.9 | 78.1 |
| LADMIM | **90.3** | 83.1 | **86.6** |

Table 8: **The effect of masking strategies.** We report image-level AUROC [%].

| Masking | SA | LA | Avg. |
|:---|:---:|:---:|:---:|
| Checkerboard | 88.1 | 78.2 | 83.2 |
| Random-0.2 | 85.1 | 75.1 | 80.1 |
| Random-0.4 | 85.3 | 76.3 | 80.8 |
| Random-0.6 | 87.2 | 78.9 | 83.1 |
| **BlockRandom-0.2** | 87.5 | **80.9** | **84.2** |
| BlockRandom-0.4 | 88.2 | 78.1 | 83.2 |
| BlockRandom-0.6 | 88.1 | 78.6 | 83.4 |
| Adaptive-0.4 | **91.0** | 77.2 | 84.1 |

Table 9: **The effect of codebook selection.** We report image-level AUROC [%] on the Box.

| Source codebooks | SA | LA | Avg. |
|:---|:---:|:---:|:---:|
| [1] | 80.1 | 78.4 | 79.3 |
| [2] | 80.7 | 78.2 | 79.5 |
| [3] | 79.0 | 77.9 | 78.5 |
| [4] | 79.5 | 78.1 | 78.8 |
| [1,2] | 82.2 | 79.3 | 80.8 |
| [1,2,3] | 82.3 | 79.0 | 80.7 |
| [1,2,3,4] | **86.2** | **81.0** | **83.6** |

Also, in Fig. 4, we plot histograms of the predicted anomaly scores produced by HVQ-Trans and ViT. Since each score has a different scale and distribution, we apply normalization before summing them. In Table 7, we evaluate detection performance with different aggregation strategies, including simple addition, min-max, and Z-score normalization. As the histograms in Fig. 4 indicate, each score distribution contains some outlier points, which are potentially anomalous images. We apply Z-score aggregation, motivated by the presence of these outliers, as Z-scores are more robust to outliers. Furthermore, for the Box category, $\mathcal{S}_{\mathrm{MIM}}$ attains the highest AUROC for logical anomaly detection. This suggests that detection accuracy could be further improved by dynamically adjusting the aggregation weights.

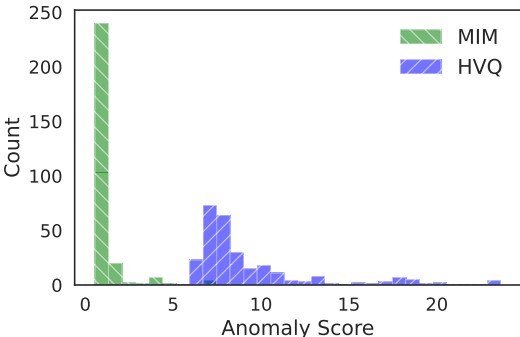

Figure 4: **Score Histogram.** We plot histograms of $\mathcal{S}_{\mathrm{HVQ}}$ and $\mathcal{S}_{\mathrm{MIM}}$ before aggregation.

Table 7: **The effect of score aggregation.** We report average detection AUROC [%] on the Box for different aggregation strategies.

| Strategies | LA | SA | Avg. |
|:---|:---:|:---:|:---:|
| $\mathcal{S}_{\mathrm{MIM}}$ | **92.0** | 63.6 | 77.8 |
| $\mathcal{S}_{\mathrm{HVQ}}$ | 79.6 | 79.5 | 79.6 |
| **Aggregation** | | | |
| addition | 82.9 | 81.1 | 82.0 |
| min-max | 85.9 | 82.5 | 84.2 |
| Z-score | 87.4 | **82.6** | **85.0** |

Table 10: **Impact of loss design.** We report image-level AUROC [%] on the `Box`.

| Metric | SA | LA | Avg. |
|---|---|---|---|
| KL divergence | 85.1 | 80.5 | 82.8 |
| JS divergence | 84.1 | 80.4 | 82.3 |
| Hellinger distance | 79.4 | 78.2 | 78.8 |
| L1 distance | **86.2** | **81.0** | **83.6** |

Table 11: **Impact of changing the MIM prediction target.** We report image-level AUROC [%]

| Prediction Target | SA | LA | Avg. |
|---|---|---|---|
| Pixel | **91.1** | 74.8 | 82.2 |
| Feature | 88.9 | **83.4** | 86.1 |
| Code | 90.7 | 78.0 | 83.9 |
| Code Histogram | 90.3 | 83.1 | **86.6** |

### 4.3.3 The Effect of Masking Strategies

In the context of MIM, the choice of masking strategies affects the properties of the learned representations Bao et al. (2021); He et al. (2021); Assran et al. (2023). Moreover, in unsupervised AD, previous work has attempted to identify the optimal masking strategies Li et al. (2020); Huang et al. (2023a); Yan et al. (2021).

In Table 8, we show the image-level AUROC [%] of LADMIM on MVTecLOCO with different masking strategies. For checkerboard masking, we randomly select one of the checkerboard masks with different grid sizes $3, 6, 12$. For random masking, we denote Random-$r$ as random masking where $r$ portion of patches are masked. For block random masking, we denote BlockRandom-$r$ as block-wise random masking where $r$ portion of patches are masked. We randomly choose the aspect ratio of each block from $[0.3, 3]$ and its position, ensuring that the sum of the gathered masks covers the $r$ portion of the patches. Finally, we also investigate an adaptive masking strategy, where masked patches are determined by the top-$r$ portion of patches with the largest prediction errors. We denote Adaptive-$r$ as this masking strategy and set $r$ to 0.4, which achieved the best performance in our hyperparameter search. For all masking strategies, we train LAViT for 200 epochs in the multi-class setting and report the average AUROC across all categories.

We observe that mask coverage is important for random masking: a higher masking ratio leads to better performance. Conversely, for block random masking, a large masking coverage results in significant uncertainty in the masked regions, causing degradation in detection performance. While checkerboard masking works well as a random masking strategy, block random masking with 20% masking shows the best detection performance in this setting. Since adaptive masking focuses the model on small potential anomalous regions, it outperforms other strategies in detecting structural anomalies. However, detection performance for logical anomalies is limited due to the overly focused mask region.

### 4.3.4 Selection of Prediction Targets

In Table 9, we report image-level detection performance on the `Box` category for different subsets of codebooks from HVQ-Trans as targets. While performance is limited when predicting only one of these codebooks, combining them yields substantial performance improvements. This is because predicting multiple histograms from different codebooks reduces the variance of the prediction targets and captures different feature granularities.

In Table 10, we also investigate the impact of MIM loss design. Notably, the simple L1 distance achieves the best detection performance, while KL or JS divergence is one of the most popular options for calculating the discrepancy between different probabilistic distributions. However, in our setting, target histograms can be sparse probabilistic vectors due to the low codebook usage. This motivates us to use the L1 distance as a loss function to improve training stability.

From the perspective of representation learning using MIM, the selection of prediction targets in masked regions is crucial Bao et al. (2021); Peng et al. (2022); Chen et al. (2024b); Assran et al. (2023). This also applies to AD, where the choice of prediction targets—such as predicting the image itself or features within the masked region—significantly affects detection performance. Table 11 shows the detection performance of ViT with different prediction targets. In this evaluation, we assessed performance when predicting the image,

features (*i.e.,* outputs of the backbone network), discrete representations, and the probability distribution of discrete representations in the masked region.

The results showed that the average detection performance was highest when using the probability distribution of discrete representations. When the image itself was the prediction target, detection performance for structural anomalies was the highest, but performance for logical anomalies was low. This is because accurate reconstruction of local image features is necessary for detecting structural anomalies, while capturing higher-level feature relationships is more important for detecting logical anomalies. Detection performance for logical anomalies improved when using discrete representations or features. However, the performance was still inferior to the proposed method due to the positional uncertainty of objects.

### 4.3.5 Analysis on Internal Representations of HVQ-Trans

Figure 5 shows a visualization of the discrete latent variable predictions made by HVQ-Trans. Here, we use a normal sample of a Juice Bottle from MVTecLOCO as an example. HVQ-Trans predicts a map of discrete latent variables for each layer, as shown in the upper part of Fig. 5.

By observing the prediction results, we can see that different discrete latent variables are predicted to some extent based on the semantics of the object. Additionally, the lower part of Fig. 5 shows image patches corresponding to specific discrete latent variables. For example, the 4th discrete latent variable in the codebook corresponds to the contents of the bottle, while the 2nd discrete latent variable corresponds to the bottle's label.

On the other hand, the predictions made by HVQ-Trans are not perfect. As shown in the lower part of Fig. 5, the fourth discrete latent variable in the codebook represents not only the contents of the bottle but also the bottle itself and the background. It predicts the same discrete latent variable for labels of different products, such as cherries and oranges. This issue, where the same discrete latent variable is predicted for different objects, is referred to as *Code Collision* Liu et al. (2023a); Hou et al. (2022); Huang et al. (2023b). Additionally, multiple different discrete latent variables can be predicted for the same object, a problem

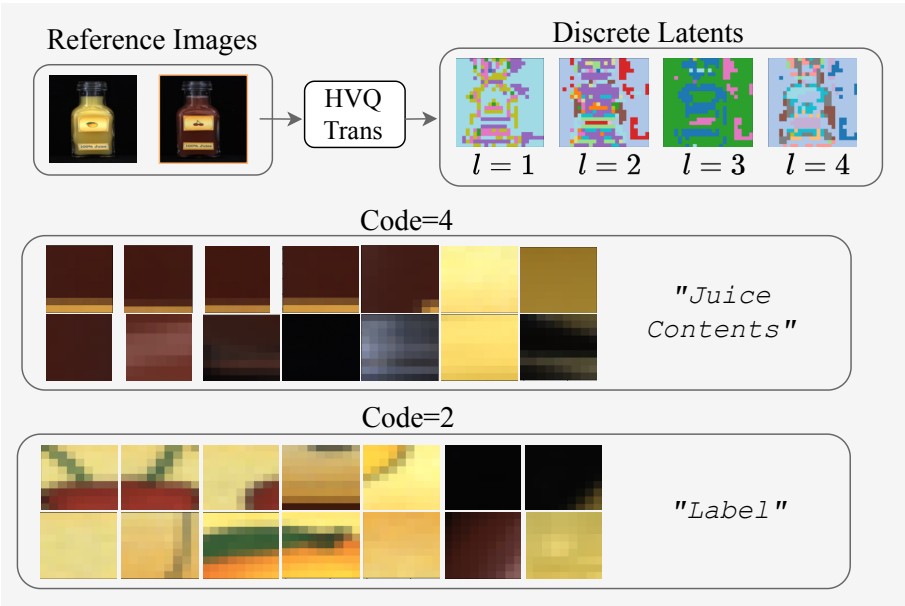

Figure 5: **Qualitative visualization** of discrete latent variables prediction by HVQ-Trans. We visualize quantized discrete latents in the $l$-th layer where $l \in \{1, 2, 3, 4\}$ in our experiments.

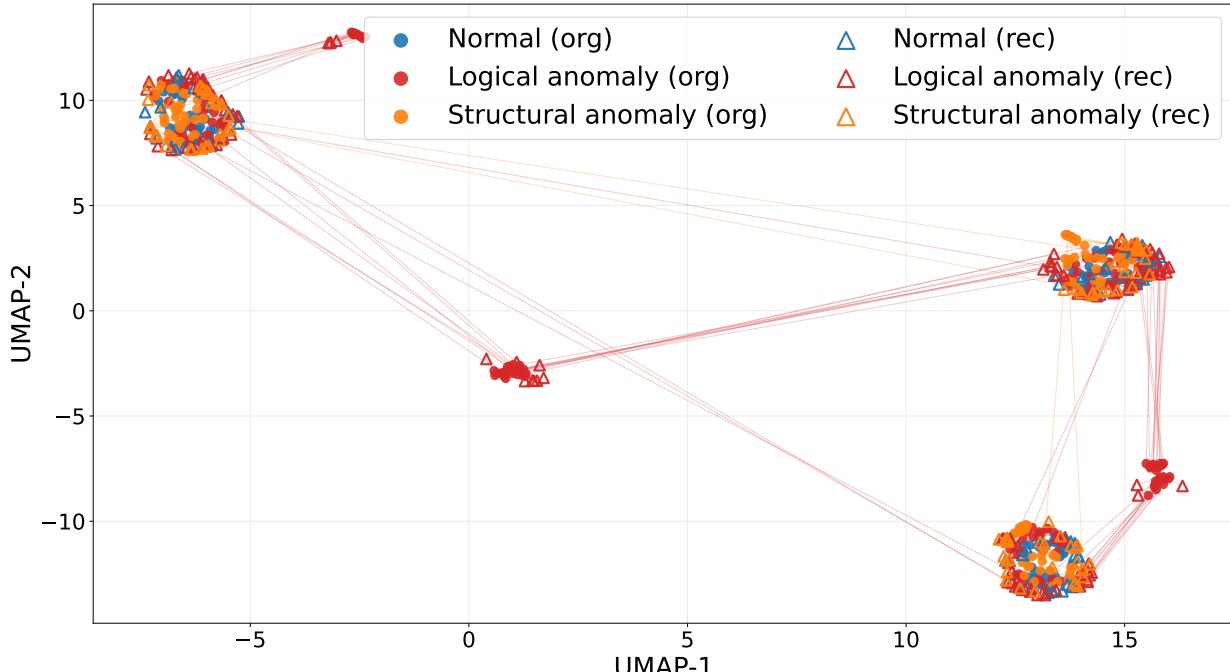

Figure 6: **Qualitative visualization** of original and reconstructed features by HVQ-Trans on `Juice Bottle`. We apply UMAP McInnes et al. (2018) to the average pooled feature representations for the dimensional reduction. UMAP-1 and UMAP-2 simply denote the two coordinates of the 2D UMAP embedding computed from pooled feature representation using cosine similarity.

known as *Code Redundancy* Huang et al. (2023c); Liu et al. (2023a); Huang et al. (2023b). Such tokenizer prediction issues significantly impact detection performance.

In Fig. 6, we visualize the distributions of the original features and their corresponding reconstructions using UMAP McInnes et al. (2018). Since UMAP reflects local neighborhood structure in the original feature space, we can easily analyze the dynamics of HVQ-Trans reconstruction in high-dimensional feature space. For visualization, we project the original features into a two-dimensional space using UMAP based on cosine similarity. Each axis (UMAP-1 and UMAP-2) is simply one coordinate of the learned 2D UMAP embedding and has no intrinsic physical meaning; the embedding can be arbitrarily rotated or reflected without changing the neighborhood relationships. It should be noted that structural anomalies are not distinguished in these visualization experiments because average pooled operation diminishes the difference between normal and anomalous patches. We observe that most logical anomalies are successfully reconstructed into near-normal features and exhibit large reconstruction errors. However, there are many logical anomalies that cannot be reconstructed into a normal one. Therefore, we still need to employ an additional MIM module for detecting such logical anomalies.

### 4.4 Failure Case Analysis

One limitation of MIM-based approaches is their sensitivity to the masking strategy. Since MIM-based methods detect anomalous regions by computing prediction errors over masked regions, the mask must effectively cover the anomalous areas. However, if the mask is too large, it introduces significant uncertainty within the masked region. Therefore, the masking strategy must strike a better balance between coverage and uncertainty.

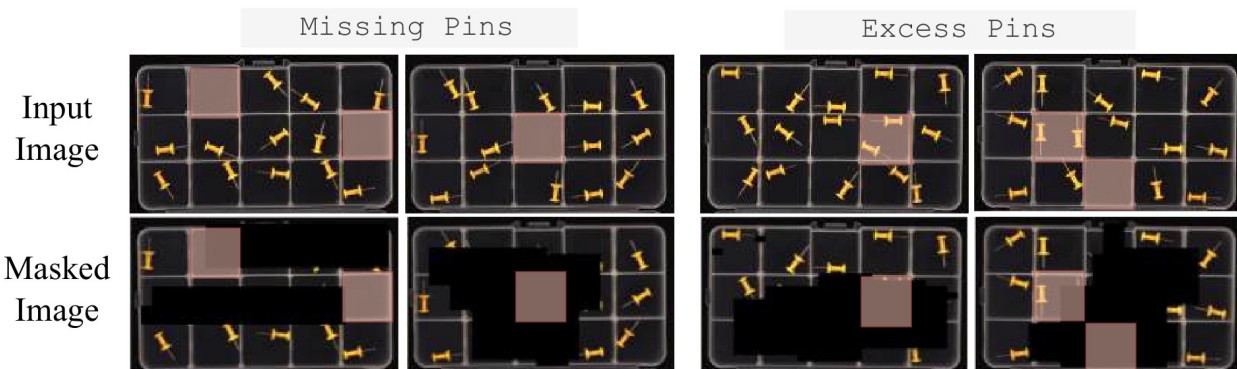

Figure 7: **Failure case analysis.** We provide examples of logical anomalies that the current SoTA method (EfficientAD) correctly detected, but LADMIM failed to detect. In the `Pins` category, each compartment in the image must contain exactly one pushpin. Ideally, each mask should cover only a single compartment; however, with random masking, this condition is often violated.

In Fig. 7, we provide examples of logical anomalies on *Pushpins* where LADMIM failed to detect anomalies but EfficientAD succeeded. If the mask is too large relative to these anomalous areas, it fails to detect missing or excessive elements. Additionally, a randomly selected mask may not adequately cover regions where anomalies are likely to occur, leading to smaller prediction errors even in anomalous areas.

Thus, a carefully designed masking strategy could boost performance in settings like *Pushpins*. For example, similar to SSM Huang et al. (2023a), iterative refinement of the mask based on reconstruction error could be considered as an inference-time masking strategy.

### 4.5 Computational Cost Analysis

In Table 12, we compare inference efficiency with frame per second (FPS) on a single RTX-4090. While Patch-Core and SALAD exhibit lower inference efficiency due to heavy feature-matching or segmentation-model computation, our proposed LADMIM achieves a second-best inference speed.

Table 12: **Inference speed** measured by frame per second (FPS) on single RTX-4090.

| Method | FPS |
| --- | --- |
| PatchCore | 13 |
| SALAD | 15 |
| EfficientAD | 260 |
| LADMIM (Ours) | 78 |

### 5 Conclusion

In this study, we addressed the problem setting of AD in which multiple objects are present in an image, and both logical and structural anomalies can occur. To effectively detect both types of anomalies, we combined a reconstruction-based model (HVQ-Trans) and a ViT trained with an MIM objective; these models are responsible for detecting structural and logical anomalies, respectively. Specifically, the ViT trained with MIM addresses the problem of positional uncertainty in masked regions by predicting distributions of discrete latents. We demonstrated substantial accuracy improvements compared to conventional MIM-based methods and comparable performance to the current SoTA methods. Additionally, we clarified the impact of changing the prediction target in the MIM-based model and highlighted issues related to the tokenizer, suggesting future directions for improving MIM-based models.

**Limitations.** While predicting code histograms mitigates the uncertainty problem in masked regions, it makes it harder to detect logical anomalies that require modeling order-related relationships, although other types of logical anomalies can still be effectively detected. This limitation can be alleviated by introducing a stochastic prediction module, such as diffusion-based masked prediction. Additionally, with the current large-scale random masking strategy, our method fails to localize the logical anomaly location. For the

future direction, replacing random masking with small-scale semantic masking would improve logical anomaly localization.

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
