# OpenReview forum: "Logical Anomaly Detection with Masked Image Modeling"
_TMLR — Accepted by TMLR_

### Review · Reviewer_9764 · 2025-12-26

**Summary Of Contributions:**

Unfortunately this paper is outside my area of expertise and thus I am unable to review it.

**Audience:**

No

**Audience Explanation:**

Unfortunately this paper is outside my area of expertise and thus I am unable to review it.

**Broader Impact Concerns:**

Unfortunately this paper is outside my area of expertise and thus I am unable to review it.

**Claims And Evidence:**

No

**Claims Explanation:**

Unfortunately this paper is outside my area of expertise and thus I am unable to review it.

**Requested Changes:**

Unfortunately this paper is outside my area of expertise and thus I am unable to review it.

---

### Review · Reviewer_nwVm · 2026-01-01

**Summary Of Contributions:**

This paper proposes LADMIM, a unified unsupervised anomaly detection framework that effectively detects both structural and logical anomalies. By combining HVQ-Trans and ViT, the method focuses on predicting the probability distribution of discrete latent variables in masked regions, which is invariant to positional uncertainty and better captures global logical relationships. Extensive experiments on MVTecLOCO and MVTecAD show that LADMIM significantly outperforms existing MIM-based approaches and achieves competitive performance with state-of-the-art methods.

**Audience:**

Yes

**Audience Explanation:**

This paper addresses the important and relatively underexplored problem of logical anomaly detection in unsupervised settings, which is relevant to the TMLR audience interested in representation learning and self-supervised methods.

**Broader Impact Concerns:**

No concerns.

**Claims And Evidence:**

Yes

**Claims Explanation:**

Strengths
* Effectively targets logical anomalies arising from object relationships, which are typically challenging for anomaly detection methods that rely primarily on local features.
* Adopts a clear functional separation between structural and logical anomaly detection, leveraging HVQ-Trans for structural anomalies and a ViT-based MIM model for logical anomalies.
* Demonstrates substantial performance improvements over existing MIM-based anomaly detection methods for both logical and structural anomaly detection tasks.
* Operates in a fully unsupervised manner without requiring segmentation priors, enhancing its applicability and generalization potential.

Weaknesses
* Is unable to detect order-sensitive logical anomalies, as the histogram-based discrete latent distribution does not preserve spatial or sequential ordering information.
* Provides limited discussion on time-based robustness and distribution shift, since the experimental evaluation is restricted to static datasets.

**Requested Changes:**

* Investigate adaptive or uncertainty-aware masking strategies at inference time. Introducing adaptive masking based on reconstruction or prediction uncertainty may improve robustness and reduce failure cases.
* Provide a more detailed analysis of computational cost and efficiency trade-offs.
* Text and numbers in the plot (such as figure 3) is small that it is sometimes hard to read. Please increase the font size to be more readable.

---

> ### Author Response · Authors · 2026-01-25
> **Response to Reviewer nwVm**
>
> ## Rebuttal to Reviewer *nwVm*
>
> First, we sincerely thank the reviewer, nwVm, for the thorough evaluation of our paper and for the valuable feedback.
> We address each comment in detail below.
>
> ---
>
> ## Weaknesses
>
> ### Q1. Order-sensitive logical anomalies
>
> > **Comment:** Is unable to detect order-sensitive logical anomalies, as the histogram-based discrete latent distribution does not preserve spatial or sequential ordering information.
>
> **A1.**
> Yes. This is a limitation of the proposed approach. We mentioned it in the **Limitations** section in **Sec. 5**.
>
> ---
>
> ### Q2. Time-based robustness and distribution shift
>
> > **Comment:** Could you provides discussion on time-based robustness and distribution shift, since the experimental evaluation is restricted to static datasets.
>
> **A2.**
> Extending our framework to handle temporal drift or non-stationary data streams would require additional modeling and evaluation protocols, which are beyond the scope of this paper. We consider this a promising avenue for future research.
>
> ---
>
> ## Requested Changes
>
> ### C1. Adaptive or uncertainty-aware masking at inference time
>
> > **Request:** Investigate adaptive or uncertainty-aware masking strategies at inference time. Introducing adaptive masking based on reconstruction or prediction uncertainty may improve robustness and reduce failure cases.
>
> **A1.**
>
> We investigate the effectiveness of an adaptive masking strategy based on MIM prediction errors. See Tab.8 in the revised manuscript. In summary, adaptive masking is effective for detecting structural anomalies, but limited for detecting logical anomalies.
>
> ---
>
> ### C2. Computational cost and efficiency trade-offs
>
> > **Request:** Provide a more detailed analysis of computational cost and efficiency trade-offs.
>
> **A2.**
>
> We provide an inference efficiency report in the Sec 4.5.  Our method achieves superior inference efficiency compared to the memorybank-based method (PatchCore) and the segmentation-based method (SALAD).
>
> ---
>
> ### C3. Improve plot readability (font size)
>
> > **Request:** Text and numbers in the plot (such as figure 3) is small that it is sometimes hard to read. Please increase the font size to be more readable.
>
> **A3.**
>
> Thank you for this comment. We agree that the text and numbers in the figures were too small and could be difficult to read.
> We have increased the font size in all figures, including Figure 3, to improve readability.
>
> ---

---

### Review · Reviewer_uPQ7 · 2026-01-12

**Summary Of Contributions:**

**Summary**
- This research aims to solve an anomaly detection task that is widely used in industry in an unsupervised manner. In particular, it focuses on the logical aspect among logical and structural anomalies and proposes Logical Anomaly Detection with Masked Image Modeling (LADMIM). By leveraging both Masked Image Modeling, which has recently made a significant impact in the ML community, and the Vector Quantization technique to perform logical anomaly detection, it shows superior performance compared to existing methods on two benchmark datasets.

**Strenghts**
- In Figure 4, the intermediate results of the network that readers may be curious about are effectively presented, and in particular, it effectively shows what kind of visual characteristics each code represents. The feature maps of the discrete latents are also presented to help understanding by showing which parts each layer focuses on.

- By introducing discrete representation learning, it demonstrates the possibility of extending the method toward a foundational model. Through this, it is expected to be able to make predictions in more diverse and complex anomaly detection environments, and additional experiments on this seem necessary.

- Including Figure 2, the methodology in Section 3 is kindly explained by dividing it into steps. The explanation is so detailed that there were no parts that were difficult to understand.


**Weaknesses**
- The logical justification for applying MIM and Vector Quantization is insufficient. While the characteristics of MIM and Vector Quantization can be understood through the related works, the paper does not include empirical results or theoretical reasons explaining why previous studies did not apply them, or why they are difficult to apply directly to this task.

- Overall, the experiments are insufficient. First, the comparison baselines are limited, and even those are not the latest studies but outdated works (2021, 2023, 2024). Many studies were published in 2025, and comparative experiments with them are necessary. In addition, the datasets are limited to only two: MVTecLOCO and MVTecAD. Comparisons in more diverse environments, including both baseline models and LADMIM, would better demonstrate the effectiveness of this work. Finally, if the results of LADMIM and the comparison models are visualized and shown differently for each case, it would make it easier for readers to understand.

**Audience:**

Yes

**Audience Explanation:**

The study “Logical Anomaly Detection with Masked Image Modeling” seems likely to provide interest to TMLR’s audience. Anomaly detection is a very traditional field and is a representative task for applying deep learning in industry. In particular, distinguishing between logical and structural anomalies and detecting them in that way makes it easier for users to interpret the results. In addition, the combination of the widely used MIM architecture and the Vector Quantization technique enables more robust predictions. Although the working mechanism of the method is not specifically visualized, for example through intermediate result examples, the quantitative results of the methodology are meaningful. It seems that experiments could be conducted on more diverse benchmark datasets and environments, and this work is expected to influence attempts to combine these techniques not only in anomaly detection but also in other areas.

**Broader Impact Concerns:**

This work does not involve privacy-sensitive personal data or ethically problematic datasets; therefore, there are no broader impact concerns to report.

**Claims And Evidence:**

No

**Claims Explanation:**

As described in the Weaknesses, the direction of this work is novel, but the process of validating that direction is insufficient. There are no experiments showing what advantages of MIM and VQ led the authors to choose them when constructing the architecture, and how, through them, the method becomes able to capture what existing approaches fail to predict. In addition, there are many variations of Vector Quantization, but the paper does not make it clear why the Hierarchical Vector Quantized Transformer was selected among numerous options. To justify this choice, the work should either provide comparisons or ablation studies with other Vector Quantization methods, or theoretically explain how the hierarchical property helps anomaly detection in the feature space; however, this cannot be confirmed in the paper. Furthermore, there are no concrete experiments showing what the MIM and VQ methodologies have learned in the feature space. As the authors claim, the paper needs to empirically demonstrate what the proposed method learns in the feature space in order for the authors’ logic to be convincing.

**Requested Changes:**

- Please explicitly mention in the text the meaning of "l" in Fig. 4. Please add it to the main body or explain it in the caption.

- A more detailed explanation of Fig. 5 is needed. Although there is an explanation in Section 4.4, assuming the readers are people who do not know this dataset and task well, it is necessary to describe the characteristics of the dataset and explain in a detailed and friendly way what each mask and result specifically mean.

- Adding experimental results by separating Unsupervised Anomaly Detection and Supervised Anomaly Detection could increase the credibility of this work. In particular, the comparative experiments are insufficient. SCADN in Table 4 is a study from as long as four years ago, and this is an outdated study in the rapidly changing deep learning community. Accordingly, I additionally request quantitative and qualitative comparisons with the latest SOTA papers such as [1], [2], [3], [4], [5]. Concrete comparisons with the above methods would be able to emphasize the technical strengths of LADMIM.

- It seems necessary to specifically mention what the ID shortcut problem is. Rather than simply citing prior work, it is necessary to explain it concretely for readers as a problem that this work aims to solve.

- I understand the intention of performing Vector Quantization. However, is there a reason for choosing the Hierarchical Vector Quantized Transformer among numerous Vector Quantization methods? Please explain what characteristics the hierarchical approach has compared to other techniques among Vector Quantization methods. Also, please describe how such advantages can be used in Logical Anomaly Detection.

- It would be good to add a logical explanation of how the simple image reconstruction error, Mean Square Error, is used as an anomaly score.

- What is the basis for claiming that the comparison methods in Table 3 have “implicit leakage of logical anomaly information”? Did that work reveal such aspects, or was it written directly by the authors of this paper?

- Although the work showed through design ablation that each core network of HVQ-Trans is robust to logical and structural anomalies, is there no way to verify this using t-SNE visualization of the network output features?

- What meaning does the architecture of this work have beyond being a combination of existing MIM methods and VQ methods?

[1] A unified anomaly synthesis strategy with gradient ascent for industrial anomaly detection and localization (ECCV 2024)

[2] Hyperbolic anomaly detection (CVPR 2024)

[3] Towards real unsupervised anomaly detection via confident meta-learning (ICCV 2025)

[4] SALAD--Semantics-Aware Logical Anomaly Detection (ICCV 2025)

[5] Towards Training-free Anomaly Detection with Vision and Language Foundation Models (CVPR 2025)

---

> ### Author Response · Authors · 2026-01-25
> **Response to Reviewer uPQ7**
>
> ## Rebuttal to Reviewer *uPQ7*
>
> We sincerely thank the reviewer, uPQ7, for the careful review of our manuscript and for recognizing the motivation and novelty of our work.
> We address each comment in detail below.
>
> ## Requested Changes
>
> ### C1. Clarify the Meaning of “l” in Fig. 4
>
> > **Request:** Please explicitly mention in the text the meaning of "l" in Fig. 4. Please add it to the main body or explain it in the caption.
>
> **A1.**
>
> Thank you for this comment. We have added a description of the meaning of "l" in the caption of Fig. 4.
>
> ---
>
> ### C2. Provide a More Reader-Friendly Explanation of Fig. 5
>
> > **Request:** A more detailed explanation of Fig. 5 is needed. Although there is an explanation in Section 4.4, assuming the readers are people who do not know this dataset and task well, it is necessary to describe the characteristics of the dataset and explain in a detailed and friendly way what each mask and result specifically mean.
>
> **A2.**
>
> Thank you for this comment. We have added a description of the logical constraints in this dataset and why our approach fails for these images in the caption of Fig.7.
>
> ---
>
> ### C3. Strengthen Comparisons with Recent SOTA Methods
>
> > **Request:** Adding experimental results by separating Unsupervised Anomaly Detection and Supervised Anomaly Detection could increase the credibility of this work. In particular, the comparative experiments are insufficient. SCADN in Table 4 is a study from as long as four years ago, and this is an outdated study in the rapidly changing deep learning community. Accordingly, I additionally request quantitative and qualitative comparisons with the latest SOTA papers such as [1], [2], [3], [4], [5]. Concrete comparisons with the above methods would be able to emphasize the technical strengths of LADMIM.
>
> **A3.**
>
> For the SOTA comparison in the MVTecAD, we have updated Tab. 4 to include the above latest methods [1,2,3]. Also, we include [4,5] in Tab.3. While [4,5] achieves superior performance than LADMIM, they still rely on a pre-trained segmentation prior.
>
> ---
>
> ### C4. Explain the ID Shortcut Problem in Detail
>
> > **Request:** It seems necessary to specifically mention what the ID shortcut problem is. Rather than simply citing prior work, it is necessary to explain it concretely for readers as a problem that this work aims to solve.
>
> **A4.**
>
> Thank you for this comment. We have added a description about the ID shortcut problem in Sec 2.1.
>
> ---
>
> ### C5. Justify Using a Hierarchical VQ Transformer for Logical Anomaly Detection
>
> > **Request:** I understand the intention of performing Vector Quantization. However, is there a reason for choosing the Hierarchical Vector Quantized Transformer among numerous Vector Quantization methods? Please explain what characteristics the hierarchical approach has compared to other techniques among Vector Quantization methods. Also, please describe how such advantages can be used in Logical Anomaly Detection.
>
> **A5.**
>
> In Sec 3.2, we have added an explanation of why we chose HVQ-Trans as a tokenizer for MIM among various VQ methods. In summary, there are two reasons: (i) HVQ-Trans achieves satisfactory detection accuracy for structural anomalies.
> (ii) HVQ is important for mitigating uncertainty in the masked region and increasing codebook usage.

---

> > ### Author Response · Authors · 2026-01-25
> > **Response to Reviewer uPQ7**
> >
> > ### C6. Explain How MSE Reconstruction Error Becomes an Anomaly Score
> >
> > > **Request:** It would be good to add a logical explanation of how the simple image reconstruction error, Mean Square Error, is used as an anomaly score.
> >
> > **A6.**
> >
> > In Sec. 3.2, we have added an explanation of why MSE is a reasonable metric for detecting structural anomalies.
> >
> > ---
> >
> > ### C7. Support the Claim of Implicit Logical Anomaly Leakage in Table 3
> >
> > > **Request:** What is the basis for claiming that the comparison methods in Table 3 have “implicit leakage of logical anomaly information”? Did that work reveal such aspects, or was it written directly by the authors of this paper?
> >
> > **A7.**
> >
> > Thank you for the comment. The statement that the comparison methods in Table 3 have “implicit leakage of logical anomaly information” was not an explicit claim made in the original papers of those methods, and our wording was misleading. To avoid unnecessary ambiguity, we have removed this statement and revised the description accordingly.
> >
> > ---
> >
> > ### C8. Validate Robustness Using t-SNE Feature Visualization
> >
> > > **Request:** Although the work showed through design ablation that each core network of HVQ-Trans is robust to logical and structural anomalies, is there no way to verify this using t-SNE visualization of the network output features?
> >
> > **A8.**
> >
> > In Fig.6 and Sec 4.3.5, we provide a visualization of input and reconstructed features of HVQ-Trans and its corresponding explanation.
> >
> > ---
> >
> > ### C9. Clarify the Novel Contribution Beyond Combining Existing Methods
> >
> > > **Request:** What meaning does the architecture of this work have beyond being a combination of existing MIM methods and VQ methods?
> >
> > **A9.**
> >
> > It should be noted that combining VQ and MIM does not yield sufficient detection accuracy for logical anomalies (LA). In Table 11, the image-level AUROC for LA using discrete codes is 78.0%, which is lower than that obtained when using continuous features as prediction targets (83.4%). One key insight of our work is that predicting code histograms improves LA detection performance (78.0% → 83.1%). To our knowledge, this is a novel idea in the context of MIM-based representation learning and prior VQ studies. Finally, we highlight this as our technical novelty in the last paragraph of Sec 3.1.

---

> > > ### Comment · Reviewer_uPQ7 · 2026-01-28
> > >
> > > Thank you for your efforts in preparing the rebuttal. Many of my initial concerns have been sufficiently addressed, and these improvements are well reflected in the revised manuscript. However, there are still several points on which I would appreciate further clarification from the authors.
> > >
> > >
> > > **C5. Justify Using a Hierarchical VQ Transformer for Logical Anomaly Detection**
> > >
> > > The authors claim that “hierarchical quantization in the HVQ-Trans increases codebook usage, which is important for subsequent MIM ViT training.” A more concrete and detailed explanation would help readers better understand this choice. Since the use of vector quantization appears to be one of the core components of the manuscript, the rationale for selecting hierarchical quantization should be elaborated more clearly. In particular, it would strengthen the argument to explain under what conditions and to what extent codebook usage increases, and to discuss the limitations of alternative vector quantization methods from the perspective of codebook utilization. Such a comparison would better justify the adoption of hierarchical quantization.
> > >
> > > **C8. Validate Robustness Using t-SNE Feature Visualization**
> > >
> > > A more explicit explanation of the UMAP visualization is required. First, please clarify what the UMAP-1 and UMAP-2 axes (x and y axes) in Figure 6 represent. In addition, the authors should explain what the distances between features indicate in this visualization and what the observed clusters correspond to. A step-by-step interpretation of the figure, including how these observations support the authors’ conclusions, would greatly improve clarity.
> > >
> > >
> > > **C9. Clarify the Novel Contribution Beyond Combining Existing Methods**
> > >
> > > In the rebuttal, the authors mainly describe their contributions in terms of performance improvements. However, I would like to see a more explicit discussion of the technical novelty. Specifically, I am interested in the design philosophy behind the selection of architectural components, the advantages of these components compared to alternative approaches, and the types of cases in which this design enables more robust predictions. Providing an explanation beyond simply stating that the proposed architecture achieves strong performance would more appropriately demonstrate the validity and significance of this research.
> > >
> > >
> > > Overall, the structure of the study and the experimental design are well-organized. However, the manuscript would benefit from a more detailed, step-by-step explanation of the methodology, with the process broken down into clearer and more explicit descriptions for the readers. If the above points are clarified accordingly, I would have no further concerns.

---

> ### Author Response · Authors · 2026-02-26
> **Official Comment by Authors**
>
> We thank the reviewer **uPQ7** for recognizing our ideas and theory as principled, as well as commending the writing and execution.
> Below, we address your questions regarding **(C5)** the choice of HVQ-Trans, **(C8)** the interpretation of Fig. 6, and **(C9)** the novelty/contribution beyond combining existing methods. We also indicate the corresponding **manuscript revisions**.
>
> ---
>
> ### C5. Justify Using a Hierarchical VQ Transformer for Logical Anomaly Detection
>
> Thank you for pointing this out. We revised the manuscript to clarify **why we choose HVQ-Trans as the MIM tokenizer**, focusing on **codebook utilization** and the mitigation of **dead codes**, which is important for subsequent MIM ViT training.
>
> ### Manuscript change
>
> * **[Sec. 3.2]** Added explanation on why HVQ-Trans is chosen as a tokenizer.
> ---
>
> ### C8. Validate Robustness Using t-SNE/UMAP Feature Visualization (Fig. 6)
>
> Thank you for the helpful suggestion. We agree that the **UMAP axes have no intrinsic physical meaning** and can be arbitrarily rotated/reflected, and that **global distances should not be over-interpreted**. We updated the manuscript to state this explicitly, and to clarify how the visualization should be read (primarily as **local neighborhood similarity** in the original feature space).
>
> ### Manuscript changes
>
> * **[Sec. 4.3.5, Fig. 6]** Added an explanation of axis meanings.
> * **[Sec. 4.3.5]** Added an explanation of what UMAP reflects.
>
> ---
>
> ### C9. Clarify the Novel Contribution Beyond Combining Existing Methods
>
> You asked specifically about the **design philosophy** behind our architectural choices, the **advantages vs. alternatives**, and the **cases where the design yields more robust prediction**. We agree this should be articulated as **technical novelty and justification**, not only performance.
>
> ### Design philosophy and technical rationale
>
> * **HVQ-Trans (tokenizer + structural AD in one module):**
>   We employ HVQ-Trans to **unify structural anomaly detection and tokenization**. It detects **structural anomalies via reconstruction error**, while simultaneously producing a **hierarchical discrete latent representation** during reconstruction. This is particularly suitable for MIM because it naturally supports **coarse-to-fine discrete targets** (multi-level granularity), which non-hierarchical VQ variants do not explicitly provide and are not designed for structural AD.
>
> * **ViT with histogram prediction (robustness to positional uncertainty):**
>   Our MIM ViT predicts the **histogram (distribution) of discrete tokens** within the masked region (instead of position-wise tokens), explicitly addressing **positional uncertainty** (e.g., local permutations or misalignment). This is important for detecting **position-invariant logical anomalies**, avoiding heuristic search procedures that can arise from an information bottleneck.
>
> * **Why MIM + VQ is justified (and what we add):**
>   Prior work (e.g., **BEiT / BEiTv2**) motivates using a VQ tokenizer for MIM because discrete tokens compress visual information into more compact targets and reduce overfitting to low-level details. Building on this foundation, our **additional technical contribution** is to extend MIM targets to **token distributions (histograms)** to handle positional ambiguity—critical in logical anomaly detection settings.
>
> ### Manuscript change
>
> * **[Sec. 3.1]** Revised the contribution statement to explicitly highlight novelty and rationale:
>
> **Reference**
>
> [1] Hangbo Bao, Li Dong, Furu Wei, “BEiT: BERT Pre-Training of Image Transformers”, *ICLR*, 2022.

---

### Review · Reviewer_i9UQ · 2026-01-15

**Summary Of Contributions:**

This work studies the problem of unsupervised logical anomaly detection in industrial inspection, where anomalies arise from incorrect global relationships among multiple objects rather than local structural defects. The authors propose LADMIM, a hybrid framework that combines a reconstruction-based model (HVQ-Trans) for structural anomaly detection with a masked image modeling (MIM) approach based on Vision Transformers for logical anomaly detection. The key idea is to predict the distribution of discrete latent variables in masked regions, rather than pixel-level or feature-level reconstructions, in order to mitigate positional uncertainty and emphasize global compositional dependencies. The method is evaluated on MVTec LOCO and MVTec AD, with extensive ablation studies.

**Audience:**

Yes

**Audience Explanation:**

The paper intersects several active research directions of interest to the TMLR community:
* Unsupervised and self-supervised anomaly detection
* Discrete latent representations and vector quantization
* Structural properties and compositionality in vision models

**Broader Impact Concerns:**

No explicit concerns.

**Claims And Evidence:**

Yes

**Claims Explanation:**

Strengths:
* The paper convincingly motivates why logical anomaly detection is fundamentally different from structural AD and why existing methods (CNN features, memory banks, pixel reconstruction) are ill-suited for capturing global dependencies. The discussion is well grounded in recent benchmarks (MVTec LOCO) and prior literature.
* The division of labor between HVQ-Trans (structural anomalies) and ViT-MIM (logical anomalies) is principled and empirically justified.
* Extensive experimental results are provided.

**Requested Changes:**

Major Comments:

* The proposed framework heavily builds upon existing components, in particular HVQ-Trans and standard MIM-based ViT architectures. While the idea of predicting histograms of discrete latent variables is interesting, the reviewer wonders whether this constitutes a fundamentally new algorithmic insight, or whether it is primarily a re-combination of existing techniques with a new interpretation tailored to logical anomaly detection.

* The core design choice—using histogram prediction in discrete latent space—removes positional and ordering information by construction. While this alleviates uncertainty in masked regions, it also fundamentally limits the types of logical anomalies that can be detected. This trade-off is only discussed later in the paper and should be more clearly acknowledged as an inherent limitation of the method.

* The MIM loss is defined as an L1 distance between predicted and target histograms. This choice is intuitive but somewhat ad hoc. The reviewer wonders if there is a specific technical reason to choose this metric rather than other distribution discrepancy criteria.

* The MIM loss sums histogram errors across all HVQ-Trans layers. It is unclear whether all layers contribute equally to logical anomaly detection. An ablation or discussion on layer weighting (or selecting a subset of layers) would improve understanding.

* The experimental evaluation, although extensive, relies exclusively on image-level AUROC. Given that the method’s performance is strongly tied to masking coverage, it is unclear how well the approach localizes anomalies or how sensitive it is to different anomaly sizes and spatial configurations. Additional evaluation beyond image-level AUROC would strengthen the empirical claims.

* The final anomaly score is obtained via a heuristic combination of standardized reconstruction and MIM-based scores. While empirically effective, this fusion lacks theoretical justification, and it is unclear how sensitive the results are to this specific choice. Also, a learned or adaptive fusion strategy could potentially improve robustness and deserves discussion.

Minor Comments:

* The relationship between $p^{\prime}$ and the probability distributions of different layers of latent variables is not clearly discussed.

---

> ### Author Response · Authors · 2026-01-25
> **Response to Reviewer i9UQ**
>
> ## Rebuttal to Reviewer *i9UQ*
>
> We appreciate Reviewer i9UQ’s detailed evaluation of our manuscript and the insightful feedback provided.
> We address each comment in detail below.
>
> ---
>
> ## Requested Changes
>
> ### C1. Assess Novelty Beyond Recombining Existing Components
>
> > **Request:** The proposed framework heavily builds upon existing components, in particular HVQ-Trans and standard MIM-based ViT architectures. While the idea of predicting histograms of discrete latent variables is interesting, the reviewer wonders whether this constitutes a fundamentally new algorithmic insight, or whether it is primarily a re-combination of existing techniques with a new interpretation tailored to logical anomaly detection.
>
> **A1.**
>
> Predicting histograms of discrete latent variables constitutes a fundamentally new algorithmic insight. To the best of our knowledge, no prior work has introduced a histogram of codes as a prediction target. While several studies have explored alternative prediction targets [1–5], they typically predict HOG (histogram of oriented gradients) features [1], discrete codes [2,3], or other feature representations [4,5], rather than a histogram of codes. Notably, our proposed prediction strategy is applicable to representation learning beyond logical anomaly detection. We therefore believe that it offers meaningful insight for the readers of TMLR.
>
> [1] Chen Wei, Haoqi Fan, Saining Xie, Chao-Yuan Wu, Alan Yuille, and Christoph Feichtenhofer, “Masked Feature Prediction for Self-Supervised Visual Pre-Training,” *Proceedings of the IEEE/CVF Conference on Computer Vision and Pattern Recognition (CVPR)*, pp. 14668–14678, June 2022.
>
> [2] Hangbo Bao, Li Dong, and Furu Wei, “BEiT: BERT Pre-Training of Image Transformers,” *arXiv preprint* arXiv:2106.08254, 2021. doi: 10.48550/arXiv.2106.08254.
>
> [3] Zhiliang Peng, Li Dong, Hangbo Bao, Qixiang Ye, and Furu Wei, “BEiT v2: Masked Image Modeling with Vector-Quantized Visual Tokenizers,” *arXiv preprint* arXiv:2208.06366, 2022. doi: 10.48550/arXiv.2208.06366.
>
> [4] Alexei Baevski, Wei-Ning Hsu, Qiantong Xu, Arun Babu, Jiatao Gu, and Michael Auli, “data2vec: A General Framework for Self-supervised Learning in Speech, Vision and Language,” *Proceedings of the 39th International Conference on Machine Learning (ICML)*, pp. 1298–1312, 2022. *(PMLR, Vol. 162)*
>
> [5] Mahmoud Assran, Quentin Duval, Ishan Misra, Piotr Bojanowski, Pascal Vincent, Michael G. Rabbat, Yann LeCun, and Nicolas Ballas, “Self-Supervised Learning from Images with a Joint-Embedding Predictive Architecture,” *Proceedings of the IEEE/CVF Conference on Computer Vision and Pattern Recognition (CVPR)*, pp. 15619–15629, 2023. doi: 10.1109/CVPR52729.2023.01499.
>
> ---
>
> ### C2. Acknowledge Limitations from Histogram-Based Order Removal
>
> > **Request:** The core design choice—using histogram prediction in discrete latent space—removes positional and ordering information by construction. While this alleviates uncertainty in masked regions, it also fundamentally limits the types of logical anomalies that can be detected. This trade-off is only discussed later in the paper and should be more clearly acknowledged as an inherent limitation of the method.
>
> **A2.**
>
> This is a limitation of the proposed approach. We have added a description of this in the Limitations section in Sec 5.
>
> ---
>
> ### C3. Justify Using L1 Distance for Histogram Loss
>
> > **Request:** The MIM loss is defined as an L1 distance between predicted and target histograms. This choice is intuitive but somewhat ad hoc. The reviewer wonders if there is a specific technical reason to choose this metric rather than other distribution discrepancy criteria.
>
> **A3.**
>
> In Tab. 10, we have investigated other popular loss functions for measuring the discrepancy between two different distributions. In summary, our loss selection is motivated by the sparsity of the target histogram, driven by relatively lower codebook usage. While L1 loss is less popular than KL/JS divergence, this metric is robust to the sparsity of the target histogram, leading to more stable training.

---

> > ### Author Response · Authors · 2026-01-25
> > **Response to Reviewer i9UQ**
> >
> > ### C4. Analyze Layer Contributions and Weighting in Multi-Layer MIM Loss
> >
> > > **Request:** The MIM loss sums histogram errors across all HVQ-Trans layers. It is unclear whether all layers contribute equally to logical anomaly detection. An ablation or discussion on layer weighting (or selecting a subset of layers) would improve understanding.
> >
> > **A4.**
> >
> > In Tab. 9, we have investigated selecting a subset of layers from HVQ-Trans and demonstrated that combining multiple layers improves detection performance. This highlights the importance of HVQ and its code diversity across layers.
> >
> > ---
> >
> > ### C5. Extend Evaluation Beyond Image-Level AUROC and Mask Sensitivity
> >
> > > **Request:** The experimental evaluation, although extensive, relies exclusively on image-level AUROC. Given that the method’s performance is strongly tied to masking coverage, it is unclear how well the approach localizes anomalies or how sensitive it is to different anomaly sizes and spatial configurations. Additional evaluation beyond image-level AUROC would strengthen the empirical claims.
> >
> > **A5.**
> >
> > Thank you for the comment. We agree that additional evaluation beyond image-level AUROC would strengthen the empirical claims. In our current setting, localizing logical anomalies with MIM-ViT is challenging due to the high masking ratio, which reduces the availability of spatial cues for precise pixel-level attribution. Consequently, localized metrics, such as pixel-level AUROC, do not exhibit a large performance gap relative to HVQ-Trans, and the overall trends are similar to those of the image-level results. This is an important limitation of our current approach, and we believe that locality-aware masking (e.g., more localized/adaptive masking) is a promising direction for improving localization. We added this discussion to Sec. 5 (Limitations).
> >
> > ---
> >
> > ### C6. Provide Rationale and Robustness Analysis for Score Fusion Strategy
> >
> > > **Request:**
> >
> > **A6.**
> >
> > In Fig. 4 and Tab. 7, we added an explanation of why we use Z-score normalization rather than simply summing them. This is because the two score distributions differ in scale and deviation. While min-max scaling is another option for normalizing them, we adopt Z-score normalization because of the presence of outliers in the AD setting.
> >
> > ---
> >
> > ### C7. Clarify the Relationship Between p' and Layer-Wise Latent Distributions
> >
> > > **Request:** The relationship between p' and the probability distributions of different layers of latent variables is not clearly discussed.
> >
> > **A7.**
> >
> > Thank you for this comment. We have already stated the explicit relationship between p' and target histograms in Eq. (10) and its surrounding parts.
> >
> > ---

---

### Author Response · Authors · 2026-01-25
**Response to the Editor and Reviewers**

We would like to express our sincere gratitude to the Action Editor, Dahun Kim, and all reviewers for their thorough assessment and valuable feedback. In particular, we greatly appreciate the Action Editor’s effort in inviting additional reviewers, which strengthened the review process and helped us improve the manuscript. We have revised the paper to address the comments and clarify the contributions.

---

### Author Response · Authors · 2026-03-13
**Acknowledgment and thanks**

Dear Editor and Reviewers,

Thank you for your careful review and for the constructive feedback on our manuscript. We greatly appreciate the time and effort you devoted to evaluating our work. Your comments and suggestions have been very helpful in improving the quality of the paper.

We are honored that our paper has been accepted and look forward to contributing to the community through this work.

Sincerely,

Authors

---

### Decision · Action_Editor_1jsZ · 2026-02-22

**Recommendation:** Accept with minor revision

**Additional Comments:**

The recommended revision mainly includes the following updates per reviewers' comments and requested changes. 1. technical novelty clarification, 2. tokenizer justification, 3. visualization context (UMAP feature Fig 6), 4. incorporate the newly requested experiments into the main text (comparison with recent SOTA baselines, evaluation of the adaptive masking strategy, loss function comparisons, computational cost analysis), 5. clearly state the limitations regarding its inability to detect order-sensitive logical anomalies and anomalies due to high masking ratios, 6. adjust visual elements in Fig 3 have a sufficiently large font size for readability as requested by the reviewers.

**Audience:**

Yes

**Audience Explanation:**

The topic is on the intersection of unsupervised anomaly detection and masked image modeling. These are active areas in the ML and TMLR community. Its focus on the distinction between structural and logical anomalies and provides practical insights for those who work on global scene understanding or industrial inspection scenarios.

**Claims And Evidence:**

Yes

**Claims Explanation:**

The paper provides extensive empirical results on standard benchmarks like MVTec LOCO and AD. The rebuttal addressed reviewers' concerns regarding baseline comparisons and technical justifications. While the initial submission lacked some ablation details for hierarchical VQ and histogram loss, the rebuttal and revised manuscript successfully clarified these design choices with new experiments.